# RETRACTED: Investigation of the Potential of Nebivolol Hydrochloride-Loaded Chitosomal Systems for Tissue Regeneration: In Vitro Characterization and In Vivo Assessment

**DOI:** 10.3390/pharmaceutics13050700

**Published:** 2021-05-11

**Authors:** Noha Ibrahim Elsherif, Abdulaziz Mohsen Al-Mahallawi, Abdelfattah Ahmed Abdelkhalek, Rehab Nabil Shamma

**Affiliations:** 1Department of Pharmaceutics and Pharmaceutical Technology, Faculty of Pharmacy, Heliopolis University, Cairo 11785, Egypt; nelsherif11@gmail.com; 2Department of Pharmaceutics and Industrial Pharmacy, Faculty of Pharmacy, Cairo University, Cairo 11562, Egypt; Abdulaziz.mohsen@pharma.cu.edu.eg; 3Department of Pharmaceutics, Faculty of Pharmacy, October University for Modern Sciences and Arts (MSA), Giza 12451, Egypt; 4Department of Microbiology of Supplementary General Science, Faculty of Oral and Dental Medicine, Future University in Egypt, Cairo 11835, Egypt; abdosc_2020@yahoo.com

**Keywords:** nebivolol hydrochloride, wound healing, tissue regeneration, chitosan lactate, chitosomes

## Abstract

In this study, we evaluated the synergistic effect of nebivolol hydrochloride (NVH), a third-generation beta-blocker and NO donor drug, and chitosan on the tissue regeneration. Ionic gelation method was selected for the preparation of NVH-loaded chitosomes using chitosan lactate and sodium tripolyphosphate. The effect of different formulation variables was studied using a full factorial design, and NVH entrapment efficiency percentages and particle size were selected as the responses. The chosen system demonstrated high entrapment efficiency (73.68 ± 3.61%), small particle size (404.05 ± 11.2 nm), and good zeta potential value (35.6 ± 0.25 mV). The best-achieved formula demonstrated spherical morphology in transmission electron microscopy and amorphization of the crystalline drug in differential scanning calorimetry and X-ray diffraction. Cell culture studies revealed a significantly higher proliferation of the fibroblasts in comparison with the drug suspensions and the blank formula. An in vivo study was conducted to compare the efficacy of the proposed formula on wound healing. The histopathological examination showed the superiority of NVH-loaded chitosomes on the wound proliferation and the non-significant difference in the collagen deposition after 15 days of the injury to that of intact skin. In conclusion, NVH-loaded chitosomes exhibited promising results in enhancing skin healing and tissue regeneration.

## 1. Introduction

Small diffusible molecules like nitric oxide (NO) play an important role in wound repair and tissue regeneration [1], as confirmed by an elevated level of NO in wounds [2,3,4]. NO has been shown to activate angiogenesis and promotes fibroplasia [5]. Recently, Nebivolol hydrochloride (NVH) has been used for its tissue regenerative abilities owing to its NO donor ability [6]. NVH is a third-generation beta-blocker approved by the FDA for the treatment of hypertension. It exerts its vasodilating effects via NO pathway by releasing cardiovascular endothelial NO [7], in addition to its conventional beta-blocking effects [8]. NVH causes endothelium dependent vasodilatation associated with the activation of L-arginine/nitric oxide pathway in both hypertensive and normotensive patients [9,10,11,12]. Being a class II drug according to the Biopharmaceutical Classification System (BCS), NVH suffers from high permeability and low solubility [13,14]. Ulger et al. conducted an experimental study to compare the effect of nebivolol to that of dexpathenol on wound healing [5]. They found that the wound healing rates of dexpanthenol, the widely used agent for wound healing was not significantly different than that of nebivolol. In addition, studies showed that NVH slows diabetic neuropathy [11], and restores endothelial function in diabetic wounds via increasing the NO supply to the treated area [11,15]. NVH also has antioxidant activity, exhibiting its effect due to the reduction of reactive oxygen species, produced by Nicotinamide Adenine Dinucleotide Phosphate (NADP) oxidase system [10,16,17].

The healing process is an interaction of complex cellular events. Thus, the presence of more than one tissue regenerative mechanism could be of a positive effect. The extracellular matrix (ECM) is an important factor in orchestrating and guiding tissue regeneration [18,19]. The ECM consists of proteins synthesized by fibroblasts, including proteoglycans (e.g., chondroitin sulfate), keratin sulfate, heparin sulfate and fibrous proteins like laminin, type IV collagen and elastin [20]. It also serves as a deposit for growth factors, proteases, cytokines, and chemokines [21]. In this view, the use of new polymers in the treatment of wounds is essential. These polymers should be biocompatible, absorbable, biodegradable, low to absent toxicity, and have no immune-stimulatory activities [20]. 

An example of these polymers is chitin and its derivative chitosan [22], gelatin [23], and hyaluronic acid [24]. Chitosan is a bioactive polymer produced from chitin, one of the most abundant natural polysaccharides globally, second only to cellulose [25,26,27]. Chitosan is composed of d-glucosamine and N-acetyl-d-glucosamine units, linked by β-1,4 glycosidic linkages [26,28] that are vulnerable to biodegradation [29,30]. It is metabolized by lysozyme, a human enzyme that breaks it down slowly to N-acetyl-β-D-glucosamine. This metabolite stimulates the main biochemical activities in wound healing, including polymorph nuclear cell activation, fibroblast activation, cytokine production, giant cell migration, and aids in regular collagen deposition in addition to stimulating hyaluronic acid synthesis at the wound site [31,32,33]. All of these functions facilitate wound contraction [34,35]. Moreover, chitosan has antibacterial properties, good biocompatibility, hemostatic and mucoadhesive powers [20,32,36,37,38]. Due to its availability and its superior advantages, it has been widely used in wound-dressing applications [35,39,40].

The aim of this study was to develop and evaluate NVH-loaded chitosomes as a potential dual function system in enhancing the wound healing process. Ionic gelation was chosen as a preparation technique to prepare NVH-loaded chitosomes via crosslinking of chitosan lactate and sodium tripolyphosphate. A full factorial design was employed to evaluate and optimize different variables. The morphology of the chitosomes prepared was evaluated using a transmission electron microscope. A cell culture study was done to evaluate the ability of the prepared chitosomes to proliferate fibroblasts. In addition, an in vivo study was done on rats, where the selected formulation was compared with the drug suspension and blank chitosomes to assess their ability to help in tissue healing and regeneration. Finally, histopathology was done for the excised skin samples to identify the formula’s ability to proliferate and reconstruct collagen.

## 2. Materials and Methods

### 2.1. Materials

Nebivolol hydrochloride (NVH) was kindly provided from Marcyrl Pharmaceutical Industries (Cairo, Egypt). Chitosan lactate (CSL) and sodium tripolyphosphate (TPP) were purchased from Sigma-Aldrich Chemical Co. (St. Louis, MI, USA). Tween 80, methanol, potassium dihydrogen phoshate (KH_2_PO_4_), disodium hydrogen phosphate (Na_2_HPO_4_), sodium chloride (NaCl) and potassium chloride (KCl) were purchased from Adwic, El-Nasr Pharmaceutical Co. (Cairo, Egypt). All other chemicals and solvents were of analytical grade and used as received. Sodium chloride intravenous infusion B.P. 2001 (normal saline) was purchased from a local pharmacy and stored according to the package information leaflet.

### 2.2. Preparation of NVH-Loaded Chitosomes Using a Full Factorial Design

NVH-loaded chitosomes were prepared using a 2^3^ full factorial design employing Design Expert^®^ software (Version 10, Stat-Ease Inc., Minneapolis, MN, USA), followed by an analysis of variance (ANOVA) to evaluate the significance of each factor [41]. The ratio of the volume of CSL to TPP solutions (X_1_), the amount of NVH (X_2_), and the concentration of CSL solution used (X_3_) were selected as the independent variables. The dependent variables (responses) were entrapment efficiency percentage EE% (Y_1_) and particle size PS (Y_2_) (Table 1), and the factor was considered significant at *p* ≤ 0.05. The preparation method was described by Berthold et al. [42] and Hashad et al. [43] with some modifications, and the composition of the prepared NVH-loaded chitosomes are shown in Table 2. An accurately weighted amount of CSL was dissolved in distilled water to create different concentrations of CSL solution. Accurately weighted amount of NVH was dissolved in methanol (4 mL) and added to the CSL solution on a magnetic stirrer (Jenway 1000, Jenway, Staffordshire, UK) at 1000 rotation per minute (rpm) at 25 °C for 1 min. After that, the calculated volume of 0.25% TPP aqueous solution was added in a dropwise manner to the mixture. The formed dispersion was stirred continuously at 1000 rpm on a magnetic stirrer for 30 min at 50 °C until the evaporation of the organic phase (methanol). The formed NVH-loaded chitosomes were then left to cool overnight at 5 °C before further evaluation.

### 2.3. Characterization of NVH-Loaded Chitosomes

#### 2.3.1. Determination of NVH Entrapment Efficiency Percentage (EE%)

To determine the efficiency of the drug encapsulation process, NVH-loaded chitosomes were assayed for entrapped drug content. Samples of 1 mL of the prepared chitosomes were centrifuged at 15,000 rpm for 1 h at 4 °C using a cooling centrifuge (Hermle Labortechnik GmbH; Wehingen, Baden-Württemberg, Germany). The supernatant was then discarded and the residue was dissolved in methanol and measured spectrophotometrically (Shimadzu, UV-1800, Kyoto, Japan) at the predetermined λ_max_ (281.7 nm) on the basis of a standard curve previously constructed [44,45]. The EE% was calculated as follows and all measurements were done in triplicates at 25 °C:
EE%=mass of entrapped NVHTotal mass of NVH × 100


#### 2.3.2. Determination of Particle Size Distribution

The PS and polydispersity index (PDI) of the NVH-loaded chitosomes were measured using a Zetasizer Nano ZS instrument (Malvern Instruments, Malvern, UK), which utilizes a light scattering technique. Each preparation was appropriately diluted using distilled water (1:10 *v/v*) to measure the PS and PDI. All measurements were done in triplicates at 25 °C [46,47,48].

#### 2.3.3. In Vitro Release of the Selected NVH-Loaded Chitosomes

The release of NVH from the selected NVH-loaded chitosomes and the NVH-aqueous suspension was conducted using the dialysis bag diffusion technique [49] in a thermostatically controlled water bath shaker (Memmert, GmbH; Buchenbach, Germany) with phosphate buffer saline (pH 7.4) containing 1% Tween 80 as the release media to ensure sink conditions, maintained at 32 ± 0.5 °C [11]. Before the experiment, the cellulose dialysis membrane (Visking^®^ dialysis tubing, diameter 21 mm, MWCO 12,000–14,000 daltons, Serva, Heidelberg, Germany) was soaked overnight in the release medium. An amount equivalent to 1 mg entrapped NVH-loaded chitosomes from the selected formulation was suspended in 1 mL of distilled water. As a control, NVH suspension (NVH suspended in distilled water at 1 mg/mL) was placed in the dialysis bag then tied at both ends. The dialysis bag was immersed in a beaker containing 100 mL of the release media and shaken using a thermostatically controlled shaker adjusted at 100 strokes per minute. At predetermined time intervals over 24 h, samples of 3 mL from the release medium were collected and assayed spectrophotometrically at the predetermined λ_max_ (283.0 nm). Each withdrawn sample was replaced by an equal volume of the fresh release medium and each experiment was done in triplicate.

#### 2.3.4. Zeta Potential of the Selected NVH-Loaded Chitosomes

Samples of 0.1 mL of the selected chitosomes were diluted into 10 mL of distilled water and measured using Zetasizer Nano-ZS (Malvern Instruments, Malvern, Worcestershire, UK) to measure the zeta potential (ZP). Each measurement was performed in triplicate at 25 °C [50,51,52], and the viscosity of the samples was assumed to be equal to that of water [53].

#### 2.3.5. Transmission Electron Microscopy (TEM) of the Selected NVH-Loaded Chitosomes

Morphological examination of the selected NVH-loaded chitosomes was carried out using a transmission electron microscope (Jeol JEM1230, Tokyo, Japan). This test was used to examine the size, sphericity and aggregation of the prepared chitosomes. The prepared NVH-loaded chitosomes were stained using 2.5% phosphotungstic acid, and then one drop of the dispersion was deposited on the surface of a carbon-coated copper grid; and allowed to dry at room temperature for 10 min before investigation by TEM [54,55]. The preparations were viewed at 15,000× and 60,000× magnifications.

#### 2.3.6. Differential Scanning Calorimetry (DSC) of the Selected NVH-Loaded Chitosomes

The thermal properties of the selected NVH-loaded chitosomes, the corresponding physical mixture, pure NVH, and CSL were evaluated using DSC (Mettler-Toledo International Inc., Columbus, OH, USA). Before conducting the DSC scanning, the selected formulation was lyophilized in order to transfer it to dry powder. The selected formulation was lyophilized by freezing the formulation at −20 °C, followed by freeze-drying for 24 h in a freeze dryer (Novalyphe-NL 500; Savant Instruments Corp., Hicksville, NY, USA) [56]. Samples of approximately 5 mg were weighed and analyzed in hermetically sealed aluminum pans for the DSC test. Samples were heated at a scanning rate of 10 °C/minute between 25–415 °C, using nitrogen as a blanket gas. The formulation excipients and the drug were used as a standard reference for comparison.

#### 2.3.7. X-ray Diffractometry (XRD) of the Selected NVH-Loaded Chitosomes

X-ray diffraction (XRD) analysis was conducted for the lyophilized selected NVH-loaded chitosomes; CSL and the corresponding physical mixture of CSL and NVH and compared with that of pure NVH. The results were recorded on an X-ray diffractometer (Scintag X-ray diffractometer, Cupertino, CA, USA) using Ni-filtered CuKα radiation at a wavelength of 1.542 A, generated with 40-kV accelerating potential and 20 mA tube current. The instrument was operated in the continuous scanning speed over a 2θ range of 5° to 50° [57].

#### 2.3.8. Effect of Storage on the Selected NVH-Loaded Chitosomes

The selected NVH-loaded chitosomes were stored at a temperature of 4 ± 2 °C and 25 ± 2 °C for a period of 6 months. Stability was assessed by comparing the appearance and results of EE% and PS before and after storage.

### 2.4. Cell Culture Study and In Vitro Evaluation of Human Fibroblast Cell Proliferation

#### 2.4.1. Sterilization of the Samples by Gamma Radiation

NVH suspension (B) along with blank selected chitosomes (C) and NVH-loaded selected chitosomes (D) was first sterilized using gamma radiation at 1 KGy for 1 h at the Egyptian Atomic Energy Authority, to ensure the sterility of the samples [39].

NVH suspension was prepared by suspending 20 mg NVH in 12 mL distilled water to correspond to the selected NVH-loaded formulation. The blank selected chitosomes were prepared using the same method described in Section 2.2 without the addition of NVH.

#### 2.4.2. HDFa Cells Culture

Human dermal fibroblasts (adult HDFa) were used in this study and were obtained from the American Type Culture Collections^®^ (ATCC, PCS-201-012™, Boulevard, Manassas, VA, USA). The cells were cultivated in Fibrolife^®^ serum-free medium for 24–27 h at 37 °C and 5% CO_2_ in T75 culture flasks (Corning^®^, New York, NY, USA).

#### 2.4.3. Effects of Samples on the HDFa Cell Viability

A determination of the percentage viable cells was done using 3-(4,5dimethyl-2-thiazolyl)-2,5-diphenyl-2H-tetrazolium bromide (MTT) assay [40]. Briefly, the selected samples and a control sample were suspended (as triplicates [58]) with HDFa cells in Fibrolife^®^ serum-free medium at concentration of 5 × 10^4^ cell/well in Corning^®^ 96-well tissue culture plates to achieve eight concentrations for each compound, then incubated for 5 days, 10 days and 15 days.

After each selected period, the number of viable cells was determined by MTT assay, where 10 μL of 12 mM MTT stock solution (5 mg MTT per 1 mL PBS pH 7.4) was added into each cell [59], followed by incubating the 96-well plates at 37 °C and 5% CO_2_ for 4 h. The cells were periodically viewed under an inverted microscope (Olympus BX63 Life Science, Tokyo, Japan) to detect the presence of intracellular punctuate purple precipitation. When the purple precipitate is clearly visual, 50 μL of DMSO was added to each well and mixed thoroughly with the pipette and incubated at 37 °C for 10 min. A microplate reader (680 XR reader, BIORAD, Hercules, CA, USA) was used at 590 nm to determine the number of viable cells via optical density. In addition, the percentage of viability was calculated using this equation:
% viability=ODtODc × 100

where *ODt* is the mean optical density of the wells treated with the tested samples, while *ODc* is the mean optical density of the control untreated cells.

#### 2.4.4. Pattern of HDFa Cell Proliferation

To determine the HDFa proliferation pattern, HDFa cells were incubated with the NVH-aqueous suspension (0.1% *w*/*v*) (B), blank selected chitosomes (C) and the selected NVH-loaded chitosomes (D) for 5, 10 and 15 days in eight-chamber cell culture slides (5 × 10^4^ cells/chamber, Life Sciences, New York, NY, USA) for each period. After that, the cells were stained with acridine orange (AO, 100 μg/mL in PBS pH 7.4), which is a nucleic acid-binding dye, and were examined by fluorescence microscopy with 62HE BFP/GFP/HcRed filter (Olympus BX63 Life Science, Tokyo, Japan) and the photos were captured using a digital camera (Olympus DP80, Tokyo, Japan). The number of viable cells was determined by green emission, and yellow and orange emissions showed indications of the fragments of nuclei and cell death [40,60]. A control group (A) with no treatment was also examined for comparison.

### 2.5. In Vivo Animal Study

#### 2.5.1. Animal Model

All animal studies were performed following the protocols approved by the Research Ethics Committee (REC, PI 1965, 27 April 2017) of Faculty of Pharmacy, Cairo University, Cairo, Egypt, and complies with the ethical guidelines and regulations of the international guiding principles for the use of animals in biomedical experiments.

Sixteen adult male and female albino Sprague-Dawley rats weighing 150–200 g were included in this study. The study subjects were randomly divided into four groups, each of four animals. The study was performed to evaluate the effect of different treatments; NVH-aqueous suspension selected NVH-loaded chitosomes and their corresponding blank chitosomes on wound healing and tissue regeneration.

Before starting the study, the animals were kept separately in polycarbonate cages to prevent any damages that might occur due to interactions between them. The temperature was kept at 25 ± 1 °C and humidity 45–55% and illuminated with artificial fluorescent light that was maintained on a 12/12 reversed light cycle. The animals were kept in four cages (4 rats per cage), with free access to food (standard diet) and water [11,40] at the Heliopolis University’s animal house. To ensure the rats healthy for the experiment, the rats were checked daily for any abnormalities. The samples used were sterilized using the same technique used before the HDFa cell culture study.

#### 2.5.2. Wound Induction Protocol

An excision wound model was used to evaluate wound closing and tissue regeneration [40,61]. For induction of wound, rats were anesthetized with thiopental sodium (25 mg/kg) [62], and then their back hair was shaved carefully using a razor. The application field was outlined with a marking pen just before creating the injury. The wound in each rat was created on the side of the spine using a sterile biopsy punch needle (No. 10, Kai Industries Co., Ltd., Seki, Japan) on the skin’s dorsal to subcutaneous in-depth in the shape of a circle with a 100mm diameter [15,63]. The wounds were left undressed to the open environment.

All wounds were cleaned with sterile normal saline daily, and then the formula tested was applied, once per day every morning, in sufficient amounts to evenly cover the entire wound area [15,61]. The first group (A) served as a control group, receiving no treatment. The second group (B) was treated with NVH suspension. The third group (C) was treated with the blank selected chitosomes, while the fourth group (D) was treated by the selected NVH-loaded chitosomes. The endpoint of this study was the complete healing of the induced wound in any of the groups [61,64], and any subject that showed formation of pus or abscess was removed from the study and reported.

#### 2.5.3. Evaluation of Wound Healing Process

The wounds were photographed using a standard mobile camera (Samsung 20 Megapixel, Suwon, Korea) at different times. Each wound area was evaluated for the presence of abscess, pus, blood and inflammation, and the area of each wound was measured using a caliper to calculate the wound size healed [40,64] as follows:
wound size %=wound size at nth dayinitial wound size×100


#### 2.5.4. Histopathological Examination of Wound Granulating Tissue

After completion of wound healing test (after 15 days), rats were sacrificed by shoulder dislocation [63]. Autopsy samples of skin containing dermis and hypodermis were isolated using sterile biopsy punch needle (No. 10, Kai Industries Co., Ltd., Seki, Japan), carefully trimmed, flushed and fixed in 10% neutral formalin solution for 72 h [65]. The samples were washed with distilled water, then processed in serial grades of ethanol, and cleared in xylene. The samples were then infiltrated and embedded into paraplast tissue embedding media. After that, the tissue sections were cut at a 4-micrometer (μm) thickness by a rotary microtome (Leica Microsystems SM2400, Cambridge, UK) for the demonstration of the skin layers in different samples. Tissue sections were stained with hematoxylin and eosin (H and E) for the study general morphological examination of the tissues [11,65,66] and Masson’s trichrome stain for the demonstration of dermal collagen fibers [11,67], and then both stained sections were examined by light microscope (Leica Microsystems GmbH, Wetzlar, Germany) [65].

#### 2.5.5. Statistical Analysis

The derived data from the HDFa cell viability, wound-healed percentage and collagen fiber measurements were statistically analyzed by SPSS^®^ version 22.0 (IBM Corp., Chicago, IL, USA) using One way ANOVA, where least square difference (LSD) test was used to determine the significance of differences at *p* < 0.05.

## 3. Results and Discussion

### 3.1. Preparation of NVH-Loaded Chitosomes

Preliminary studies were performed to carefully select the different parameters that might affect the preparation of these vesicles, such as chitosan type and molecular weight, degree of chitosan acetylation, chitosan concentration, TPP concentration and chitosan:TPP mass ratio [68,69]. Various chitosan types were used, including chitosan with low molecular weight, chitosan with high molecular weight, and chitosan lactate (CSL); however, CSL produced the most physically stable formulations. Ionic gelation method as described by Berthold et al. was the method of choice for preparing NVH-loaded chitosomes, as they reported the preparation of chitosan–TPP complex by adding chitosan acidic solution into a TPP solution in a dropwise manner [42]. This process depends on the use of complexation between positively charged chitosan and negatively charged TPP by electrostatic forces [70]. Neither harmful chemicals nor critical operations are required in this process and it is very simple and mild [70]; thus, this method is one of the most common methods described for the formation of chitosomes in the literature [43,71,72,73].

### 3.2. Characterization of NVH-Loaded Chitosomes

#### 3.2.1. Effect of Formulation and Process Variables on EE% (Y_1_)

The average EE% of NVH in different chitosomes ranged between 49.13 ± 0.84% and 91.50 ± 1.36% (Table 2). Results of ANOVA test showed that only increasing the amount of NVH (X_2_) had a significant effect (*p* = 0.0009) on the EE%, as shown in Figure 1A (presented as averages of the parameters). It was observed that increasing the amount of NVH from 10 mg to 20 mg lead to a significant increase in the EE% of the prepared chitosomes. This could be attributed to the possible hydrogen bond formation between NVH and CSL that could improve the EE% of the formed chitosomal nanovesicles [52]. Similar results were shared by Woraphatphadung et al. in their study on the effect of the addition of curcumin to chitosan-based pH-sensitive polymeric micelles for colon-targeting. Their results reported that increasing the amount of curcumin (from 5% to 40% in comparison to polymer) lead to an increase in the entrapment efficiency [74].

#### 3.2.2. Effect of Formulation and Process Variables on Particle Size Distribution

The PS and PDI of the prepared NVH-loaded chitosomes are shown in Table 2. PS was determined as z-average as it represents the mean hydrodynamic diameter of the particles [75,76]. PDI is used to measure the width of the dispersion of particle distribution, with numbers varying between 0–1. PDI values from 0.01 to 0.5 indicate monodispersed population and homogeneity in the particle size distribution [77], while a larger PDI reflects a higher heterogeneity. The higher the polydispersity, the lower the uniformity of the vesicle size in the formulation [78]. The PS and PDI fluctuated between 404.05 ± 11.243 and 1262.5 ± 68.58 nm, and 0.305 ± 0.047 and 0.555 ± 0.043, respectively.

Statistical analysis using ANOVA showed that both NVH amount (X_2_) (*p* = 0.0248) and CSL concentration (X_3_) (*p* = 0.0001) had a significant impact on the PS. The analysis showed that the increase in the amount of NVH led to the increase in the PS of the formed chitosomes significantly as illustrated in Figure 1B. This could be due to the increase in the EE% of the formulations containing higher drug amount. A possible explanation is that the increase in the encapsulated drug amount leads to a less solid matrix structure of CSL/TPP, leading to an increase in the PS. Mahmoud et al. shared similar results in their study of chitosan/sulfobutylether-β-cyclodextrin nanoparticles. They stated that there was a correlation between the PS and drug content, whereby an increase in the drug content values yielded an increase in the PS [79].

This was also observed by Ustundag-Okur et al. in their study on the modification of solid lipid nanoparticles containing NVH using polyethylene glycol and CSL. They found that increasing the amount of NVH lead to a significant increase in the observed particle dimensions [80]. Woraphatphadung et al. also observed the same phenomena in their study of chitosan-based pH sensitive polymeric micelles. They stated that by adding curcumin to the blank micelles led to an increase in the PS, and by increasing the amount of curcumin in the preparation, the PS also increased [74].

Concerning the CSL concentration (X_3_), it was observed that by increasing its amount, the PS of the chitosomes increase as shown in Figure 1C. This could be due to the increase in the concentration and the intrinsic viscosity of the polymer in a solution, leading to more entanglement. Sreekumar et al. studied the parameters that influence the size of chitosan-TPP nano- and microparticles. They observed that the increase in chitosan concentrations lead to an elevation in the average hydrodynamic diameter of the particles. They attributed this to the volume occupied by chitosan in a solution. They stated that the concentration and the intrinsic viscosity of the polymer in a solution are given by the volume occupied. In the low concentration solutions, the polymer coils are free to move, only generating minor frictional forces. As the concentration increases, they start to touch each other and may enter the entanglement phase, leading to a larger particle size [81]. Ustundag-Okur et al. revealed similar results upon increasing the amount of CSL in the prepared solid lipid nanoparticles formulations of NVH [80].

Based on the obtained results, the conditions for the selection of the best-achieved formulation as stated in Table 1 (achieving highest EE% and lowest PS) were found in NVH-loaded chitosomes F6 (it showed the highest desirability value of 0.795). Therefore, it was chosen for further investigations.

### 3.3. Characterization of the Best Achieved NVH-Loaded Chitosomes

#### 3.3.1. In Vitro Release Study

The in vitro release study for the best achieved NVH-loaded chitosomes was done to evaluate the release behavior of NVH from the chitosomes in comparison with the NVH-aqueous suspension.

Figure 2 represents the release profile of NVH from NVH-loaded chitosomes (F6) and NVH-aqueous suspension. The release study showed a gradual release of NVH from the loaded chitosomal nanovesicles, in comparison to the drug suspension; however, there was no significant difference in the release profile itself. The gradual release of the sample could be attributed to the retainment of NVH in the formed nanovesicles, leading to a slower release [11]. These results were shared by Jana et al. in their study of NVH nanoparticles prepared using Eudragit^®^ RS100. They found that the release of NVH-aqueous suspension was very fast (≈90% after 4 h), in comparison with their prepared nanoparticles. They attributed it to the influence of the drug-polymer complexation, leading to a slower rate of release [82].

#### 3.3.2. Zeta Potential Determination

Zeta potential (ZP) is very useful as a measurement of the overall charges acquired by the nanovesicles, and could be used to evaluate the stability of the dispersion [83,84]. The formulation is believed to be more stable when ZP value is more than ±30 mV, due to electrical repulsion between particles [85]. The value of the ZP of the best achieved NVH-loaded chitosomes (F6) was measured and found to be 35.6 ± 0.25 mV. The high positive charge could be attributed to the cationic nature of CSL [86]. Similar results were observed by Luo et al. [87] and Ustundag-Okur et al. [80] due to the presence of chitosan. In addition, the presence of NVH gives a positive charge effect on the particles, which was also observed by Ustundag-Okur et al. [80].

#### 3.3.3. Transmission Electron Microscopy (TEM)

The best achieved NVH-loaded chitosomal nanovesicles (F6) were morphologically examined using transmission electron microscope, and the photomicrograph of F6 is shown in Figure 3. The TEM micrograph showed that the developed NVH-loaded chitosomes were unilamellar, with uniform, spherical discrete shape and no fusion. The diameter of the chitosomes was very close to that observed using the Zetasizer.

#### 3.3.4. Differential Scanning Calorimetry (DSC)

DSC is a tool used to investigate the physical status of NVH and determine its nature within the developed formulation, and elucidate any possible interactions with other ingredients [80,88]. Figure 4 shows the thermograms of NVH, CSL, NVH:CSL physical mixture [similar to the optimal sample in *w*/*w*] (2:5) and the lyophilized NVH-loaded chitosomes (F6).

The DSC scan of pure NVH exhibited a melting endothermic peak at 228 °C, corresponding to its melting point [89,90]. CSL exhibited broad spectrum endothermic peaks in the temperature ranges of 70–120 °C corresponding to polymeric dehydration of the typical polysaccharide and 215–225 °C corresponding to hydrogen-bonded lactate dimers within the CSL matrix [91]. Similar behavior was reported to CSL by Cervera et al. [92] and Parize et al. [93]. An observed decrease in both peaks of NVH and CSL in the NVH:CSL physical mixture, with no disappearance of their individual characteristic peaks owing to their dilution upon mixing.

The lyophilized formulation F6 depicted diminished distinctive peak of the NVH in the DSC thermogram. This decrease in the melting point and endothermic peak could be due to formation of amorphous regions in which the drug is located, associated with numerous lattice defects [88,94]. These results suggest that the drug have been homogeneously dispersed throughout the nanovesicular formulation in an amorphous state.

#### 3.3.5. X-Ray Diffractometry (XRD)

The XRD study was done to evaluate the physical nature of the drug and the effect of the method of preparation and excipients [95,96]. Figure 5 presents the XRD of NVH, CSL, NVH:CSL physical mixture (2:5) and the lyophilized best-achieved formulation F6. NVH had sharp intense peaks at 5.9°, 11.9°, 12.2°, 16.3°, 18.4°, 21.4°, 22.4°, and 25.67°, confirming its crystalline form [95,96,97]. The X-ray spectra of the physical mixture showed that the NVH peaks intensity decreased, possibly due to the dilution effect, without a qualitative disparity of its diffractogram.

Upon the incorporation of NVH in the chitosome formula, NVH peaks diminished, proving a reduction in its crystallinity. In other words, the diminishing of certain drug peaks in the XRD of the formulation, compared with the NVH:CSL physical mixture, and NVH pure drug could indicate the conversion of NVH from the crystalline state to the amorphous state in the optimal NVH-loaded chitosomes (F6). This could be due to the encapsulation of NVH inside the optimal formula in amorphous form [95,98].

#### 3.3.6. Effect of Storage

Table 3 shows the effect of storage at two different temperatures (4 ± 2 °C and 25 ± 2 °C) for six months. It was observed that there was no significant difference (*p* > 0.05) in the examined factors after six months in comparison to those freshly prepared. This could be attributed to the highly positive zeta potential that prevented the particles from aggregation [83,85].

### 3.4. Cell Culture Study and In Vitro Evaluation of Human Fibroblast Cell Proliferation

The samples were sterilized before use, using gamma radiations, as it is a cold method with no temperature increase; thus, no harm is expected to affect the samples [99,100]. The selection of this low intensity (1 kGy) was to avoid the changes in the physio-chemical characters of the chitosomes, according to previously collected data. Morsi et al. used gamma radiations to sterilize chitosan hydrogel for using in Saos-2 cell line viability [39]. Their results showed that a dose of 10 kGy had nearly neglectable effects on their formulation characteristics. Desai and Park have studied the gamma-radiation effects on chitosan microparticles [101]. They observed that the thermograms of irradiated samples at up to 25 kGy were almost the same as that of the non-irradiated samples, indicating that the radiation did not alter the matrix composition. Yang et al. have studied the effect of gamma radiation on chitosan membranes performance modification [102]. They compared the infrared spectra of chitosan membranes with and without irradiation, and they observed that the irradiated chitosan membranes at 12, 14, 16 and 18 kGy had almost identical infrared spectra to those without irradiation, indicating no chemical group formation by gamma radiations. In general, the in vitro cytotoxicity testing of biomaterials and samples is used as toxic chemicals affect the basic functions and proliferation of cells thus, cellular damage is an indication for toxicity [103].

This test was used to present insights about the performance of NVH and chitosomes on HDFa proliferation, using MTT test, as fibroblast proliferation is of huge importance on wound healing and tissue regeneration [40]. The images of fluorescence microscope (Figure 6) demonstrated the HDFa cells increase in number and proliferation at 5, 10 and 15 days compared to the control group (A). It was observed that there was a systemic increase in all groups, while group (D) showed significant increase (*p* < 0.05) in fibroblast proliferation in comparison with the group treated with NVH-aqueous suspension (B) and the group treated with the blank selected chitosomes (C). This could be attributed to the synergistic effect of both chitosan and NVH on HDFa proliferation.

The results of the MTT assay showed a high viability percentage of the three groups (Figure 7), indicating the biocompatibility and non-toxic nature of the tested groups [104]. NVH-loaded chitosomes F6 (D) showed an increase in HDFa cell proliferation after 15 days in comparison with control group (A), NVH suspension (B) and blank selected chitosomes (C). It was also observed that group (B) receiving NVH-aqueous suspension had a higher proliferation of HDFa cells in comparison with the control group at all times. This could be due to NO presence from NVH, which acts as a chemo-attractant for fibroblasts and promotes fibroplasia [105].

### 3.5. In Vivo Animal Studies

#### 3.5.1. Assessment of Wound Healing

The changes in the size of wounds were assessed on the 5th, 10th and 15th day and photo images are presented in Figure 8. The wound size percentages are graphically represented in Figure 9. No signs of inflammation was observed in all the groups, and blood, pus and abscess were absent throughout the study.

The excisional wound model was selected in the current study as it is considered to resemble acute clinical wounds [64]. This model allows the investigation of the wound healing steps: inflammation, granulation, reepithelialization, angiogenesis and remodeling. The advantage of this technique is its relative simplicity and practicality. The wound bed can be easily accessed for the treatments’ application and wound assessment [106,107]. On the other hand, the wounds in the classic excisional wounds, especially in mice, will heal primarily by contraction [64], thus the need for the close monitoring of the wound healing rate by imaging [108], followed by the histopathology studies to ensure the complete tissue regeneration [109].

Throughout the study, significant differences in the improvement of the wound status in the treated groups with NVH and chitosan from the untreated group were observed. The group treated with NVH-loaded chitosomes F6 (D) showed a significantly higher contraction of the wound in comparison with the untreated group (A) after 5 days (*p* = 0.003), 10 days (*p* = 1.2 × 10^−6^) and 15 days (*p* = 4.7 × 10^−8^). After 15 days, complete closing of the wound in the NVH-loaded chitosomes treated group (D) was observed, while the wound size was reduced to 27.08 ± 2.4% of the initial wound size in the control untreated group (A). Group (D) treated with NVH-loaded chitosomes F6 showed a non-significant higher contraction of the wound compared to both groups treated with NVH suspension (B) and blank optimal chitosomes (C) after 5 days. However, it became significantly higher after the 10th day (B *p* = 0.008 and C *p* = 0.012) and the 15th day (*p* = 2.58 × 10^−6^ and *p* = 0.0006 for groups B and C respectively). These differences demonstrate the synergistic effect of both NVH and chitosan together in the formula in comparison to their singular use.

Hernandez Martinez et al. developed a nanocomposite based on chitosan containing gold-calreticulin, for wound healing and evaluated it on a diabetic mice model [110]. They observed that the developed nanocomposite was able to increase the proliferation and migration of fibroblasts favoring the process of wound healing due to presence of chitosan [111,112]. In addition, Djekic et al. developed a chitosan composite hydrogel with sustained-release ibuprofen for advanced wound dressing, which showed promising results on wound healing [113].

#### 3.5.2. Histopathological Study

Light microscopic examination of the histopathological sections of the rats stained with H and E and Masson’s trichrome for the four groups after 15 days of the wound induction and a normal control sample are illustrated in Figure 10. The untreated group (A) (Figure 10A) showed persistence of epidermal loss and ulceration as presented even after 14 days of treatment, with adjacent sub epidermal hemorrhagic patches and high cellular granulation tissue formation rich with inflammatory cells infiltrates filling the wound gap and non-organized dermal collagen deposition (dashed arrow) which was significantly (*p* = 2.28 × 10^−13^) lower than the normal control sample, indicating an incomplete process of healing [40].

Concerning the treated groups, the group treated daily with NVH suspension (B) showed still persistence focal epidermal loss and incomplete re-epithelialization as presented in Figure 10B with the black arrow with focal subepidermal hemorrhagic patches. The wound gap was replaced with cellular granulation tissue with mild inflammatory cells infiltrates and minimal organized collagen fibers formation as presented with the dashed arrow (*p* = 5.25 × 10^−8^). This could be due to the presence of NVH, and resulting in a high concentration of NO causing endothelium-dependent vasodilatation, leading to stimulation of inflammatory cells and proliferation and closure of wound at a faster rate [11]. Group C (Figure 10C) treated with the blank chitosomes showed complete epidermal re-epithelization and wound closure as presented by the black arrow, also accompanied with focal subepidermal hemorrhagic patches. There were more collagen-rich dermal granulation tissue samples presented by the dashed arrows (*p* = 6.5 × 10^−5^), with high cellular infiltrates and newly formed blood vessels (red arrow) due to the presence of chitosan [58,111,112,114].

On the other hand, the microscopical examination of the group treated with NVH-loaded chitosomes F6 (D) [depicted in Figure 10D] revealed a more accelerated wound healing process with complete epidermal re-epithelization as presented by the black arrow. The wound gap was reduced and filled with fibrous granulation tissue and high records of mature collagen bundles (dashed arrow), which was not significantly (*p* = 0.243) different from the collagen bundles present in the normal control group (Figure 10E) represented by the star, showing its superiority in aiding the wound healing and tissue regeneration.

## 4. Conclusions

In this study, NVH-loaded chitosomal nanovesicles were successfully prepared using an ionic gelation method according to a 2^3^ full factorial design. The best achieved chitosomal formula had a high EE%, high positive ZP and small PS with spherical non-aggregated morphology. Several tests were done to ensure the amorphization of NVH in those nanovesicles. The results confirmed the potential of the dual-action system (NVH and CSL) in enhancing the proliferation of fibroblasts in cell culture studies and accelerated wound healing in vivo rat model. Thus, the prepared NVH-loaded chitosomes offer a promising system for accelerated wound healing and tissue reconstruction.

## Figures and Tables

**Figure 1 pharmaceutics-13-00700-f001:** Line chart showing the effect of: (**A**) the amount of NVH (X_2_) on the EE% of NVH in NVH-loaded chitosomal formulations; (**B**) the amount of NVH (X_2_) on the PS of NVH in NVH-loaded chitosomal formulations; (**C**) the concentration of CSL (X_3_) on the PS of NVH in NVH-loaded chitosomal formulations.

**Figure 2 pharmaceutics-13-00700-f002:** Release profile of NVH from (**a**) optimal chitosomal formulation F6; (**b**) NVH suspension in water.

**Figure 3 pharmaceutics-13-00700-f003:** TEM photomicrograph of the best achieved chitosomal formulation (F6).

**Figure 4 pharmaceutics-13-00700-f004:** DSC thermograms of (**a**) physical mixture (5:2) of CSL and NVH; (**b**) lyophilized best achieved NVH-loaded chitosomal formulation F6; (**c**) CSL; (**d**) NVH.

**Figure 5 pharmaceutics-13-00700-f005:** X-ray diffractogram of (**a**) CSL; (**b**) lyophilized optimal chitosomal formulation F6; (**c**) physical mixture (5:2) of CSL and NVH; (**d**) NVH.

**Figure 6 pharmaceutics-13-00700-f006:** Fluorescence microscope images (×100) for HDFa proliferation of HDFa cells treated with: (**B**) NVH-suspension; (**C**) blank chitosomal formulation, and (**D**) NVH-loaded chitosomal formulation F6 after 5 days, 10 days and 15 days of incubation, and control (**A**) after 15 days.

**Figure 7 pharmaceutics-13-00700-f007:** The effect of NVH suspension (**B**), blank chitosomal formulation (**C**) and NVH-loaded chitosomal formulation F6 (**D**) on the proliferation of HDFa cells by MTT assay.

**Figure 8 pharmaceutics-13-00700-f008:** Photographs of the changes in the wound sizes at day 0, 5, 10 and 15, for the non-treated control group (**A**), treated group with NVH suspension (**B**), treated group with blank chitosomal formulation (**C**) and treated group with NVH-loaded chitosomal formulation F6 (**D**).

**Figure 9 pharmaceutics-13-00700-f009:** Changes of wound sizes throughout the 15 days of wound healing in non-treated control group (**A**), treated group with NVH suspension (**B**), treated group with blank chitosomal formulation (**C**) and treated group with NVH-loaded chitosomal formulation F6 (**D**).

**Figure 10 pharmaceutics-13-00700-f010:** Histopathological microscopical examination section of the non-treated control group (**A**); treated group with NVH suspension (**B**); treated group with blank chitosomal formulation (**C**) and treated group with NVH-loaded chitosomal formulation (**D**), and normal control sample skin (**E**).

**Table 1 pharmaceutics-13-00700-t001:** The independent and dependent variables for the full factorial design used for preparing NVH-loaded chitosomal systems.

Factors (Independent Variables)	Levels
X_1_: Ratio of the volume of CSL to TPP solutions	5:1	10:1
X_2_: Amount of NVH (mg)	10	20
X_3_: Concentration of CSL solution used (%)	0.5	1.5
**Responses (Dependent Variables)**	**Constraints**
Y_1_: Entrapment efficiency (%)	Maximum
Y_2_: Particle size (nm)	Minimum

**Table 2 pharmaceutics-13-00700-t002:** Formulations of the experimental design and their response results.

Formula	Independent Variables	Dependent Variables
X_1_:Ratio of the Volume of CSL Solution to TPP Solution	X_2_:Amount of NVH (mg)	X_3_:Concentration of CSL Solution Used (%)	Y_1_: EE% *	Y_2_: PS (nm) *	PDI *
F1	5:1	10	0.5	60.69 ± 3.64	640.20 ± 9.89	0.555 ± 0.043
F2	5:1	20	1.5	77.50 ± 5.22	1262.5 ± 68.59	0.451 ± 0.017
F3	10:1	10	0.5	51.20 ± 2.33	508.00 ± 18.24	0.406 ± 0.058
F4	10:1	20	1.5	65.17 ± 4.67	918.20 ± 59.84	0.419 ± 0.069
F5	5:1	10	1.5	49.13 ± 0.84	611.15 ± 42.35	0.361 ± 0.067
F6	5:1	20	0.5	73.68 ± 3.61	404.05 ± 11.24	0.479 ± 0.055
F7	10:1	10	1.5	68.61 ± 2.55	858.20 ± 58.68	0.305 ± 0.047
F8	10:1	20	0.5	91.50 ± 1.36	549.70 ± 12.44	0.511 ± 0.027

* Data are represented as mean (*n* = 3) ± S.D.

**Table 3 pharmaceutics-13-00700-t003:** Effect of storage on the selected NVH-loaded chitosomes.

Parameter	At 0 Time	After 6 Months at 4 ± 2 °C	After 6 Months at 25 ± 2 °C
EE% *	73.68 ± 3.61%	72.12 ± 4.65%	72.62 ± 0.14%
PS *	404.5 ± 11.24 nm	440.3 ± 19.31 nm	425.6 ± 21.60 nm

* Data are represented as mean (*n* = 3) ± S.D.

## Data Availability

Not applicable.

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
