# Peer review of "Investigation of the Potential of Nebivolol Hydrochloride-Loaded Chitosomal Systems for Tissue Regeneration: In Vitro Characterization and In Vivo Assessment"

_pharmaceutics, 2021, doi:10.3390/pharmaceutics13050700_

Round 1
Reviewer 1 Report
The manuscript deals with the fabrication of chitosan-based nanocarriers via a simple gelation approach for the delivery of nebivolol hydrochloride drug. The work is sure of great interest from both the academic and industrial points of view. The manuscript is well-written, the Introduction section is comprehensively presenting the topic state-of-art, the methods are described in detail, and the discussion is supported by the experimental results. Thereby, I would be willing to suggest the article for publication in Pharmacautics journal after a minor revision.
Below, a detailed list of the points that should be addressed by the Authors is reported.
- The Section of the Materials and Methods paragraph should be reorganized. It is difficult to follow the order of the characterization performed by the Authors.
- Lines 178-180
Why did the Authors use such an “unusual” sterilization method? Even if there is no heat involved, the high energy of gamma radiation can most likely induce some modifications in the sample physical-chemical properties.
- Section 3.1
This Section should be included in the Introduction section since it is not related to results presented in the present work.
- Please replace Figure 1 with high-definition images.
- Lines 396-397
The Authors should explain their findings.
Author Response
The manuscript deals with the fabrication of chitosan-based nanocarriers via a simple gelation approach for the delivery of nebivolol hydrochloride drug. The work is sure of great interest from both the academic and industrial points of view. The manuscript is well-written, the Introduction section is comprehensively presenting the topic state-of-art, the methods are described in detail, and the discussion is supported by the experimental results. Thereby, I would be willing to suggest the article for publication in Pharmaceutics journal after a minor revision.
Response: The authors thank the reviewer for his/her encouraging comments.
Below, a detailed list of the points that should be addressed by the Authors is reported.
- The Section of the Materials and Methods paragraph should be reorganized. It is difficult to follow the order of the characterization performed by the Authors.
Response: The reviewer’s comment has been followed and accordingly the order of characterization tests has been changed to be easier to follow.
- Lines 178-180
Why did the Authors use such an “unusual” sterilization method? Even if there is no heat involved, the high energy of gamma radiation can most likely induce some modifications in the sample physical-chemical properties.
Response: The authors respect the reviewer’s point of view. However, the authors want to clarify that the selection of such a low intensity gamma irradiation (1 kGy) was to avoid the changes in the physio-chemical characters of the chitosomes according to previous collected data. Morsi et al. used gamma radiations to sterilize chitosan hydrogel for using in Saos-2 cell line viability [1]. Their results showed that a dose of 10 kGy had nearly neglectable effects on their formulation characteristics. Desai and Park have studied the gamma-radiation effects on chitosan microparticles [2]. They observed that the thermograms of irradiated samples at up to 25 kGy were almost the same as that of the non-irradiated samples, indicating that the radiation did not alter the matrix composition. Also Yang et al. have studied the effect of gamma radiation on chitosan membranes performance modification[3]. They compared the infrared spectra of chitosan membranes with and without irradiation, and they observed that the irradiated chitosan membranes at 12, 14, 16 and 18 kGy had almost identical infrared spectra to that without irradiation, indicating no chemical group formation by gamma radiations. These facts has been highlighted and added to the manuscript.
- Section 3.1
This Section should be included in the Introduction section since it is not related to results presented in the present work.
Response: The authors appreciate the reviewer’s opinion. However, the authors want to clarify that this section was included in the results and discussion because it discusses the reason for choosing the applied preparation method and mentions the preliminary studies conducted to select the parameters chosen.
- Please replace Figure 1 with high-definition images.
Response: The figure has been replaced as per reviewer’s instruction.
- Lines 396-397
The Authors should explain their findings.
Response: The reviewer’s comment has been followed. Accordingly, the following paragraph has been added to results and discussion section and highlighted in the manuscript:
Chitosan lactate (CSL) exhibited broad spectrum endothermic peaks in the temperature ranges of 70-120◦C corresponding to polymeric dehydration of the typical polysaccharide and 215-225◦C corresponding to hydrogen bonded lactate dimers within the CSL matrix [4]. Similar behavior was reported to CSL by Cervera et al.[5] and Parize et al.[6]”.
Reviewer 2 report (Round 1)
Open Review
(x) I would not like to sign my review report
( ) I would like to sign my review report
English language and style
( ) Extensive editing of English language and style required
( ) Moderate English changes required
( ) English language and style are fine/minor spell check required
(x) I don't feel qualified to judge about the English language and style
Yes |
Can be improved |
Must be improved |
Not applicable |
|
Does the introduction provide sufficient background and include all relevant references? |
( ) |
( ) |
(x) |
( ) |
Is the research design appropriate? |
(x) |
( ) |
( ) |
( ) |
Are the methods adequately described? |
( ) |
( ) |
(x) |
( ) |
Are the results clearly presented? |
( ) |
(x) |
( ) |
( ) |
Are the conclusions supported by the results? |
(x) |
( ) |
( ) |
( ) |
Comments and Suggestions for Authors
The paper entitled “Investigation of the potential of nebivolol hydrochloride loaded chitosomal systems for tissue regeneration: In-vitro characterization and in-vivo assessment” includes potentially relevant data for the pharmacology of wound repair and regeneration process. The Authors of this publication describe the effect of different formulation variables studied using a full factorial design, and nebivolol hydrochloride entrapment efficiency with the used percentages and particle size.
However, I have a few remarks:
- The Authors described (in the “Introduction”section) the active pharmaceutical substance (nebivolol hydrochloride(NVH)) used in the study, however I did not find the convincing justification the choice of the implemented API – NVH, including its advantages and disadvantages;
Response: The use of NVH was mainly selected due to its NO donor ability. The importance of NO presence in the wound healing and tissue regeneration process has been reported recently. Although NO is produced endogenously, several patients might suffer from slower wound closures due to lower amounts of NO in the wound. NVH, an antihypertensive drug, has been recently used in microspheres for diabetic neuropathy treatment and in gels for diabetic foot ulcer treatment. Also an experimental study was conducted to compare the effect of dexpanthol, a widely used wound healing agent, and NVH for wound closures, and they concluded that there was no significant difference between the two treatments. All these evidence had lead to our choice of NVH for this study, and accordingly the introduction order was changed to reflect this meaning and highlighted.
- The Authors described (in the Introduction section) the innovative system of modified drug release (chitosomal systems), however I did not find the convincing justification the choice, as well as its advantages and disadvantages;
Response: The main idea for using chitosan and formulation of chitosomes was to benefit from the advantages of using chitosan itself. Chitosan is biocompatible, biodegradable, abundant, relatively cheap, non-toxic and absorbable. It also resembles in its structure the extracellular matrix that guides the tissue regeneration. In our study, we tried to entrap NVH in chitosomes to try to help NVH penetrate the skin layers, in addition to helping in the formation of ECM, and accordingly the introduction order was changed to reflect this meaning and highlighted.
- The Authors did not justify – convincingly – the choice of the implemented in the study animal model, as well as its advantages and disadvantages;
Response: The excisional wound model was selected in the current study as it is considered to resemble acute clinical wounds [46]. This model allows the investigation of the wound healing steps: inflammation, granulation, reepithelialization, angiogenesis and remodeling. The advantage of this technique is its relative simplicity and practicality. The wound bed can be easily accessed for the treatments’ application and wound assessment [47, 48]. On the other hand, the wounds in the classic excisional wounds especially in mice will heal primarily by contraction [46], thus the need for the close monitoring of the wound healing rate by imaging [49], followed by the histopathology studies to ensure the complete tissue regeneration [50]. These facts has been highlighted and added to the manuscript.
- The resolution of the figures [1. (A), 1. (B)] is insufficient – figures should be redrawn.
Response: The figures have been replaced as per reviewer’s suggestion.
- The authors should add, at least brief, information concerning the standardization of the “Wound induction protocol”.
Response:
The authors thank the reviewer for his/her comment, and a standardization of the a protocol was performed with respect to the following items [51]:
- Known and identified animal proposed for use: Albino Sprague-Dawley rats
- Known and characterized challenge agent: The study was performed to evaluate the effect of different treatments; NVH-aqueous suspension, selected NVH-loaded chitosomes and their corresponding blank chitosomes on the wound healing and tissue regeneration
- Procedural information for the challenge agent exposure: An excision wound model was used to evaluate wound closing and tissue regeneration [45, 52]. For induction of wound, rats were anesthetized with thiopental sodium (25 mg/kg)[53], and then their back hair was shaved carefully using a razor. The application field was outlined with marking pen just before creating the injury. The wound in each rat was created on the side of the spine using a sterile biopsy punch needle (No. 10, Kai Industries Co., Ltd, Seki City, Japan) on skin dorsal to subcutaneous in depth in the shape of a circle having 10 mm diameter [21, 54]. The wounds were left undressed to the open environment.
- Identification of the primary and secondary endpoints: The end point of this study was the complete healing of the induced wound in any of the groups [46, 52]
- Potential triggers for intervention: any subject that showed formation of pus or abscess was removed from the study and reported.
References:
- Morsi, N., Shamma, RN, Eladawy, NO and Abdelkhalek, AA, Bioactive injectable triple acting thermosensitive hydrogel enriched with nano-hydroxyapatite for bone regeneration: in-vitro characterization, Saos-2 cell line cell viability and osteogenic markers evaluation. Drug Dev Ind Pharm, 2019. 45(5): p. 787-804.
- Desai, K., Park, HJ, Study of gamma-irradiation effects on chitosan microparticles. Drug Deliv, 2006. 13(1): p. 39-50.
- Yang, F., Li, X, Cheng, M, Gong, Y, Zhao, N, Zhang, X, Yang, Y, Performance modification of chitosan membranes induced by gamma irradiation. J Biomater Appl, 2002. 16(3): p. 215-26.
- Al Bakain, R., Abulateefeh, SR, Taha, MO, Synthesis and characterization of chitosan-lactate–phthalate and evaluation of the corresponding zinc- and aluminum-crosslinked beads as potential controlled release matrices. European Polymer Journal, 2015. 73: p. 402-412.
- Cervera, M., Heinämäki, J, de la Paz, N, López, O, Maunu, SL, Virtanen, T, Hatanpää, T, Antikainen, O, Nogueira, A, Fundora, J and Yliruusi, J, Effects of spray drying on physicochemical properties of chitosan acid salts. AAPS PharmSciTech, 2011. 12(2): p. 637-649.
- Parize, A., Cristina, T, Souza, R, Brighente, II, Fávere, V, Spinelli, A and Longo, E, Microencapsulation of the natural urucum pigment with chitosan by spray drying in different solvents. African Journal of Biotechnology, 2010. 7.
- Frank, S., Kämpfer, H, Wetzler, C and Pfeilschifter, J, Nitric oxide drives skin repair: Novel functions of an established mediator. Kidney International, 2002. 61(3): p. 882-888.
- Luo, J., Chen, A, Nitric oxide: a newly discovered function on wound healing. Acta Pharmacologica Sinica, 2005. 26(3): p. 259-264.
- Schaffer, M., Tantry, U, Gross, SS, Wasserburg, HL and Barbul, A, Nitric oxide regulates wound healing. J Surg Res, 1996. 63(1): p. 237-40.
- Schäffer, M., Tantry, U, Ahrendt, GM, Wasserkrug, HL and Barbul, A, Acute protein-calorie malnutrition impairs wound healing: a possible role of decreased wound nitric oxide synthesis. J Am Coll Surg, 1997. 184(1): p. 37-43.
- Ulger, B., Kapan, M, Uslukaya, O, Bozdag, Z, Turkoglu, A, Alabalık, U and Onder, A, Comparing the effects of nebivolol and dexpanthenol on wound healing: an experimental study. International Wound Journal, 2016. 13(3): p. 367-371.
- !!! INVALID CITATION !!!
- Jatav, V., Saggu, JS, Sharma, AK, Sharma, A and Jat, RK, Design, development and permeation studies of nebivolol hydrochloride from novel matrix type transdermal patches. Adv Biomed Res, 2013. 2: p. 62.
- Hilas, O., Ezzo, D, Nebivolol (bystolic), a novel Beta blocker for hypertension. P & T : a peer-reviewed journal for formulary management, 2009. 34(4): p. 188-192.
- Toker, A., Gulcan, E, Toker, S, Erbilen, E and Aksakalli, E, Nebivolol Might be Beneficial in Osteoporosis Treatment: A Hypothesis. Tropical Journal of Pharmaceutical Research (ISSN: 1596-5996) Vol 8 Num 2, 2009. 8.
- Weber, M., The role of the new beta-blockers in treating cardiovascular disease. Am J Hypertens, 2005. 18(12 Pt 2): p. 169S-176S.
- Pandit, A., Patel, SA, Bhanushali, VP, Kulkarni, VS and Kakad, VD, Nebivolol-Loaded Microsponge Gel for Healing of Diabetic Wound. AAPS PharmSciTech, 2017. 18(3): p. 846-854.
- Kalinowski, L., Dobrucki, LW, Szczepanska-Konkel, M, Jankowski, M, Martyniec, L, Angielski, S and Malinski, T, Third-generation beta-blockers stimulate nitric oxide release from endothelial cells through ATP efflux: a novel mechanism for antihypertensive action. Circulation, 2003. 107(21): p. 2747-52.
- Thadkala, K., Sailu, C and Aukunuru, J, Formulation, optimization and evaluation of oral nanosuspension tablets of nebivolol hydrochloride for enhancement of dissoluton rate. Der Pharmacia Lettre, 2015. 7: p. 71-84.
- Al-Dhubiab, B., Nair, A, Kumria, R, Attimarad, M and Harsha, S, Development and evaluation of nebivolol hydrochloride nanocrystals impregnated buccal film. Farmacia, 2019. 67: p. 282-289.
- Gulcan, E., Kuçuk, A, Çayci, K,Tosun, M, Emre, H, Koral, L, Aktan, Y and Avsar, U, Topical effects of nebivolol on wounds in diabetic rats. Eur J Pharm Sci, 2012. 47(2): p. 451-5.
- Mollnau, H., Schulz, E, Daiber, A, Baldus, S, Oelze, M, August, M, Wendt, M, Walter, U, Geiger, C, Agrawal, R, Kleschyov, AL, Meinertz, T and Münzel, T, Nebivolol prevents vascular NOS III uncoupling in experimental hyperlipidemia and inhibits NADPH oxidase activity in inflammatory cells. Arterioscler Thromb Vasc Biol, 2003. 23(4): p. 615-21.
- Cosentino, F., Bonetti, S, Rehorik, R, Eto, M, Werner-Felmayer, G, Volpe, M and Lüscher, T, Nitric-oxide-mediated relaxations in salt-induced hypertension: Effect of chronic β1-selective receptor blockade. Journal of hypertension, 2002. 20: p. 421-8.
- Muñoz, G.a.Z., F, Chitosan, Chitosan Derivatives and their Biomedical Applications, 2017. p. 87-106.
- Azuma, K., Izumi, R, Osaki, T, Ifuku, S, Morimoto, M, Saimoto, H, Minami, S and Okamoto, Y, Chitin, chitosan, and its derivatives for wound healing: old and new materials. J Funct Biomater, 2015. 6(1): p. 104-42.
- Patrulea, V., Ostafe, V, Borchard, G and Jordan, O, Chitosan as a starting material for wound healing applications. Eur J Pharm Biopharm, 2015. 97(Pt B): p. 417-26.
- Hynes, R.O., The Extracellular Matrix: Not Just Pretty Fibrils. Science, 2009. 326(5957): p. 1216-1219.
- Muzzarelli, R., Chitins and chitosans for the repair of wounded skin, nerve, cartilage and bone. Carbohydrate Polymers, 2009. 76(2): p. 167-182.
- Ulubayram, K., Nur Cakar, A, Korkusuz, P, Ertan, C and Hasirci, N, EGF containing gelatin-based wound dressings. Biomaterials, 2001. 22(11): p. 1345-56.
- Estes, J., Scott Adzick, N, Harrison, MR, Longaker, MT and Stern, R, Hyaluronate metabolism undergoes and ontogenic transition during fetal development: Implications for Scar-free wound healing. Journal of Pediatric Surgery, 1993. 28(10): p. 1227-1231.
- Priyadarshi, R.a.R., J, Chitosan-based biodegradable functional films for food packaging applications. Innovative Food Science & Emerging Technologies, 2020. 62: p. 102346.
- Muxika, A., Etxabide, A, Uranga, J, Guerrero, P and de la Caba, K, Chitosan as a bioactive polymer: Processing, properties and applications. Int J Biol Macromol, 2017. 105(Pt 2): p. 1358-1368.
- Watanabe, J., Iwamoto, S and Ichikawa, S, Entrapment of some compounds into biocompatible nano-sized particles and their releasing properties. Colloids Surf B Biointerfaces, 2005. 42(2): p. 141-6.
- Younes, I., Rinaudo, M, Chitin and chitosan preparation from marine sources. Structure, properties and applications. Mar Drugs, 2015. 13(3): p. 1133-74.
- Bharadwaz, A., Jayasuriya, AC, Recent trends in the application of widely used natural and synthetic polymer nanocomposites in bone tissue regeneration. Materials Science and Engineering: C, 2020. 110: p. 110698.
- Kavya, K., Jayakumar, R, Nair, S and Chennazhi, KP, Fabrication and characterization of chitosan/gelatin/nSiO2 composite scaffold for bone tissue engineering. Int J Biol Macromol, 2013. 59: p. 255-63.
- Mezzana, P., Clinical efficacy of a new chitin nanofibrils-based gel in wound healing. Acta Chir Plast, 2008. 50(3): p. 81-4.
- Muzzarelli, R., Mattioli-Belmonte, M, Pugnaloni, A and Biagini, G, Biochemistry, histology and clinical uses of chitins and chitosans in wound healing. EXS, 1999. 87: p. 251-64.
- Sashiwa, H., Saimoto, H, Shigemasa, Y, Ogawa, R and Tokura, S, Lysozyme susceptibility of partially deacetylated chitin. Int J Biol Macromol, 1990. 12(5): p. 295-6.
- Jayakumar, R., Prabaharan, M, Sudheesh Kumar, PT, Nair, SV and Tamura, H, Biomaterials based on chitin and chitosan in wound dressing applications. Biotechnol Adv, 2011. 29(3): p. 322-37.
- Hamedi, H., Moradi, S, Hudson, SM and Tonelli, AE, Chitosan based hydrogels and their applications for drug delivery in wound dressings: A review. Carbohydr Polym, 2018. 199: p. 445-460.
- Anal, A., Stevens, W, Chitosan-alginate multilayer beads for controlled release of ampicillin. Int J Pharm, 2005. 290(1-2): p. 45-54.
- Anal, A., Tobiassen, A, Flanagan, J and Singh, H, Preparation and characterization of nanoparticles formed by chitosan-caseinate interactions. Colloids Surf B Biointerfaces, 2008. 64(1): p. 104-10.
- Berger, J., Reist, M, Mayer, JM, Felt, O, Peppas, NA and Gurny, R, Structure and interactions in covalently and ionically crosslinked chitosan hydrogels for biomedical applications. Eur J Pharm Biopharm, 2004. 57(1): p. 19-34.
- Maged, A., Abdelkhalek, AA, Mahmoud, AA, Salah, S, Ammar, MM and Ghorab, MM, Mesenchymal stem cells associated with chitosan scaffolds loaded with rosuvastatin to improve wound healing. Eur J Pharm Sci, 2019. 127: p. 185-198.
- Masson-Meyers, D., Andrade, TAM, Caetano, GF, Guimaraes, FR, Leite, MN, Leite, SN, Frade, MAC, Experimental models and methods for cutaneous wound healing assessment. Int J Exp Pathol, 2020. 101(1-2): p. 21-37.
- Wong, V., Sorkin, M, Glotzbach, JP, Longaker, MT, Gurtner, GC, Surgical approaches to create murine models of human wound healing. J Biomed Biotechnol, 2011. 2011: p. 969618.
- Peplow, P., Chung, TY, Baxter, GD, Laser photobiomodulation of proliferation of cells in culture: a review of human and animal studies. Photomed Laser Surg, 2010. 28 Suppl 1: p. S3-40.
- Papier, A., Peres, MR, Bobrow, M, Bhatia, A, The digital imaging system and dermatology. Int J Dermatol, 2000. 39(8): p. 561-75.
- Geer, D., Swartz, DD, Andreadis, ST, In vivo model of wound healing based on transplanted tissue-engineered skin. Tissue Eng, 2004. 10(7-8): p. 1006-17.
- Ibeh, B., Experimental Animal Models of Human Diseases - An Effective Therapeutic Strategynull2018: IntechOpen.
- Nagar, H., Srivastava, AK, Srivastava, R, Kurmi, ML, Chandel, HS, Ranawat, MS, Pharmacological Investigation of the Wound Healing Activity of <i>Cestrum nocturnum</i> (L.) Ointment in Wistar Albino Rats. Journal of Pharmaceutics, 2016. 2016: p. 9249040.
- Sirohi, B., Sagar, R, Effect of Hydroalcoholic Extract of Dactylorhiza Hatagirea Roots & Lavandula Stoechas Flower on Thiopental Sodium Induced Hypnosis in Mice. Journal of Drug Delivery and Therapeutics, 2019. 9: p. 414-417.
- El-Bahy, A., Aboulmagd, YM and Zaki, M, Diabetex: A novel approach for diabetic wound healing. Life Sci, 2018. 207: p. 332-339.

Reviewer 2 Report
The paper entitled “Investigation of the potential of nebivolol hydrochloride loaded chitosomal systems for tissue regeneration: In-vitro characterization and in-vivo assessment” includes potentially relevant data for the pharmacology of wound repair and regeneration process. The Authors of this publication describe the effect of different formulation variables studied using a full factorial design, and nebivolol hydrochloride entrapment efficiency with the used percentages and particle size.
However, I have a few remarks:
- The Authors described (in the “Introduction” section) the active pharmaceutical substance (nebivolol hydrochloride (NVH)) used in the study, however I did not find the convincing justification the choice of theimplemented API – NVH, including its advantages and disadvantages;
- The Authors described (in the Introduction section) the innovative system of modified drug release (chitosomal systems), however I did not find the convincing justification the choice, as well as its advantages and disadvantages;
- The Authors did not justify – convincingly – the choice of the implemented in the study animal model, as well as its advantages and disadvantages;
- The resolution of the figures [1. (A), 1. (B)] is insufficient – figures should be redrawn.
- The authors should add, at least brief, information concerning the standardization of the “Wound induction protocol”.
Author Response
Reviewer 2 report (Round 1)
Open Review
(x) I would not like to sign my review report
( ) I would like to sign my review report
English language and style
( ) Extensive editing of English language and style required
( ) Moderate English changes required
( ) English language and style are fine/minor spell check required
(x) I don't feel qualified to judge about the English language and style
Yes |
Can be improved |
Must be improved |
Not applicable |
|
Does the introduction provide sufficient background and include all relevant references? |
( ) |
( ) |
(x) |
( ) |
Is the research design appropriate? |
(x) |
( ) |
( ) |
( ) |
Are the methods adequately described? |
( ) |
( ) |
(x) |
( ) |
Are the results clearly presented? |
( ) |
(x) |
( ) |
( ) |
Are the conclusions supported by the results? |
(x) |
( ) |
( ) |
( ) |
Comments and Suggestions for Authors
The paper entitled “Investigation of the potential of nebivolol hydrochloride loaded chitosomal systems for tissue regeneration: In-vitro characterization and in-vivo assessment” includes potentially relevant data for the pharmacology of wound repair and regeneration process. The Authors of this publication describe the effect of different formulation variables studied using a full factorial design, and nebivolol hydrochloride entrapment efficiency with the used percentages and particle size.
However, I have a few remarks:
- The Authors described (in the “Introduction”section) the active pharmaceutical substance (nebivolol hydrochloride(NVH)) used in the study, however I did not find the convincing justification the choice of the implemented API – NVH, including its advantages and disadvantages;
Response: The use of NVH was mainly selected due to its NO donor ability. The importance of NO presence in the wound healing and tissue regeneration process has been reported recently. Although NO is produced endogenously, several patients might suffer from slower wound closures due to lower amounts of NO in the wound. NVH, an antihypertensive drug, has been recently used in microspheres for diabetic neuropathy treatment and in gels for diabetic foot ulcer treatment. Also an experimental study was conducted to compare the effect of dexpanthol, a widely used wound healing agent, and NVH for wound closures, and they concluded that there was no significant difference between the two treatments. All these evidence had lead to our choice of NVH for this study, and accordingly the introduction order was changed to reflect this meaning and highlighted.
- The Authors described (in the Introduction section) the innovative system of modified drug release (chitosomal systems), however I did not find the convincing justification the choice, as well as its advantages and disadvantages;
Response: The main idea for using chitosan and formulation of chitosomes was to benefit from the advantages of using chitosan itself. Chitosan is biocompatible, biodegradable, abundant, relatively cheap, non-toxic and absorbable. It also resembles in its structure the extracellular matrix that guides the tissue regeneration. In our study, we tried to entrap NVH in chitosomes to try to help NVH penetrate the skin layers, in addition to helping in the formation of ECM, and accordingly the introduction order was changed to reflect this meaning and highlighted.
- The Authors did not justify – convincingly – the choice of the implemented in the study animal model, as well as its advantages and disadvantages;
Response: The excisional wound model was selected in the current study as it is considered to resemble acute clinical wounds [46]. This model allows the investigation of the wound healing steps: inflammation, granulation, reepithelialization, angiogenesis and remodeling. The advantage of this technique is its relative simplicity and practicality. The wound bed can be easily accessed for the treatments’ application and wound assessment [47, 48]. On the other hand, the wounds in the classic excisional wounds especially in mice will heal primarily by contraction [46], thus the need for the close monitoring of the wound healing rate by imaging [49], followed by the histopathology studies to ensure the complete tissue regeneration [50]. These facts has been highlighted and added to the manuscript.
- The resolution of the figures [1. (A), 1. (B)] is insufficient – figures should be redrawn.
Response: The figures have been replaced as per reviewer’s suggestion.
- The authors should add, at least brief, information concerning the standardization of the “Wound induction protocol”.
Response:
The authors thank the reviewer for his/her comment, and a standardization of the a protocol was performed with respect to the following items [51]:
- Known and identified animal proposed for use: Albino Sprague-Dawley rats
- Known and characterized challenge agent: The study was performed to evaluate the effect of different treatments; NVH-aqueous suspension, selected NVH-loaded chitosomes and their corresponding blank chitosomes on the wound healing and tissue regeneration
- Procedural information for the challenge agent exposure: An excision wound model was used to evaluate wound closing and tissue regeneration [45, 52]. For induction of wound, rats were anesthetized with thiopental sodium (25 mg/kg)[53], and then their back hair was shaved carefully using a razor. The application field was outlined with marking pen just before creating the injury. The wound in each rat was created on the side of the spine using a sterile biopsy punch needle (No. 10, Kai Industries Co., Ltd, Seki City, Japan) on skin dorsal to subcutaneous in depth in the shape of a circle having 10 mm diameter [21, 54]. The wounds were left undressed to the open environment.
- Identification of the primary and secondary endpoints: The end point of this study was the complete healing of the induced wound in any of the groups [46, 52]
- Potential triggers for intervention: any subject that showed formation of pus or abscess was removed from the study and reported.
References:
- Morsi, N., Shamma, RN, Eladawy, NO and Abdelkhalek, AA, Bioactive injectable triple acting thermosensitive hydrogel enriched with nano-hydroxyapatite for bone regeneration: in-vitro characterization, Saos-2 cell line cell viability and osteogenic markers evaluation. Drug Dev Ind Pharm, 2019. 45(5): p. 787-804.
- Desai, K., Park, HJ, Study of gamma-irradiation effects on chitosan microparticles. Drug Deliv, 2006. 13(1): p. 39-50.
- Yang, F., Li, X, Cheng, M, Gong, Y, Zhao, N, Zhang, X, Yang, Y, Performance modification of chitosan membranes induced by gamma irradiation. J Biomater Appl, 2002. 16(3): p. 215-26.
- Al Bakain, R., Abulateefeh, SR, Taha, MO, Synthesis and characterization of chitosan-lactate–phthalate and evaluation of the corresponding zinc- and aluminum-crosslinked beads as potential controlled release matrices. European Polymer Journal, 2015. 73: p. 402-412.
- Cervera, M., Heinämäki, J, de la Paz, N, López, O, Maunu, SL, Virtanen, T, Hatanpää, T, Antikainen, O, Nogueira, A, Fundora, J and Yliruusi, J, Effects of spray drying on physicochemical properties of chitosan acid salts. AAPS PharmSciTech, 2011. 12(2): p. 637-649.
- Parize, A., Cristina, T, Souza, R, Brighente, II, Fávere, V, Spinelli, A and Longo, E, Microencapsulation of the natural urucum pigment with chitosan by spray drying in different solvents. African Journal of Biotechnology, 2010. 7.
- Frank, S., Kämpfer, H, Wetzler, C and Pfeilschifter, J, Nitric oxide drives skin repair: Novel functions of an established mediator. Kidney International, 2002. 61(3): p. 882-888.
- Luo, J., Chen, A, Nitric oxide: a newly discovered function on wound healing. Acta Pharmacologica Sinica, 2005. 26(3): p. 259-264.
- Schaffer, M., Tantry, U, Gross, SS, Wasserburg, HL and Barbul, A, Nitric oxide regulates wound healing. J Surg Res, 1996. 63(1): p. 237-40.
- Schäffer, M., Tantry, U, Ahrendt, GM, Wasserkrug, HL and Barbul, A, Acute protein-calorie malnutrition impairs wound healing: a possible role of decreased wound nitric oxide synthesis. J Am Coll Surg, 1997. 184(1): p. 37-43.
- Ulger, B., Kapan, M, Uslukaya, O, Bozdag, Z, Turkoglu, A, Alabalık, U and Onder, A, Comparing the effects of nebivolol and dexpanthenol on wound healing: an experimental study. International Wound Journal, 2016. 13(3): p. 367-371.
- !!! INVALID CITATION !!!
- Jatav, V., Saggu, JS, Sharma, AK, Sharma, A and Jat, RK, Design, development and permeation studies of nebivolol hydrochloride from novel matrix type transdermal patches. Adv Biomed Res, 2013. 2: p. 62.
- Hilas, O., Ezzo, D, Nebivolol (bystolic), a novel Beta blocker for hypertension. P & T : a peer-reviewed journal for formulary management, 2009. 34(4): p. 188-192.
- Toker, A., Gulcan, E, Toker, S, Erbilen, E and Aksakalli, E, Nebivolol Might be Beneficial in Osteoporosis Treatment: A Hypothesis. Tropical Journal of Pharmaceutical Research (ISSN: 1596-5996) Vol 8 Num 2, 2009. 8.
- Weber, M., The role of the new beta-blockers in treating cardiovascular disease. Am J Hypertens, 2005. 18(12 Pt 2): p. 169S-176S.
- Pandit, A., Patel, SA, Bhanushali, VP, Kulkarni, VS and Kakad, VD, Nebivolol-Loaded Microsponge Gel for Healing of Diabetic Wound. AAPS PharmSciTech, 2017. 18(3): p. 846-854.
- Kalinowski, L., Dobrucki, LW, Szczepanska-Konkel, M, Jankowski, M, Martyniec, L, Angielski, S and Malinski, T, Third-generation beta-blockers stimulate nitric oxide release from endothelial cells through ATP efflux: a novel mechanism for antihypertensive action. Circulation, 2003. 107(21): p. 2747-52.
- Thadkala, K., Sailu, C and Aukunuru, J, Formulation, optimization and evaluation of oral nanosuspension tablets of nebivolol hydrochloride for enhancement of dissoluton rate. Der Pharmacia Lettre, 2015. 7: p. 71-84.
- Al-Dhubiab, B., Nair, A, Kumria, R, Attimarad, M and Harsha, S, Development and evaluation of nebivolol hydrochloride nanocrystals impregnated buccal film. Farmacia, 2019. 67: p. 282-289.
- Gulcan, E., Kuçuk, A, Çayci, K,Tosun, M, Emre, H, Koral, L, Aktan, Y and Avsar, U, Topical effects of nebivolol on wounds in diabetic rats. Eur J Pharm Sci, 2012. 47(2): p. 451-5.
- Mollnau, H., Schulz, E, Daiber, A, Baldus, S, Oelze, M, August, M, Wendt, M, Walter, U, Geiger, C, Agrawal, R, Kleschyov, AL, Meinertz, T and Münzel, T, Nebivolol prevents vascular NOS III uncoupling in experimental hyperlipidemia and inhibits NADPH oxidase activity in inflammatory cells. Arterioscler Thromb Vasc Biol, 2003. 23(4): p. 615-21.
- Cosentino, F., Bonetti, S, Rehorik, R, Eto, M, Werner-Felmayer, G, Volpe, M and Lüscher, T, Nitric-oxide-mediated relaxations in salt-induced hypertension: Effect of chronic β1-selective receptor blockade. Journal of hypertension, 2002. 20: p. 421-8.
- Muñoz, G.a.Z., F, Chitosan, Chitosan Derivatives and their Biomedical Applications, 2017. p. 87-106.
- Azuma, K., Izumi, R, Osaki, T, Ifuku, S, Morimoto, M, Saimoto, H, Minami, S and Okamoto, Y, Chitin, chitosan, and its derivatives for wound healing: old and new materials. J Funct Biomater, 2015. 6(1): p. 104-42.
- Patrulea, V., Ostafe, V, Borchard, G and Jordan, O, Chitosan as a starting material for wound healing applications. Eur J Pharm Biopharm, 2015. 97(Pt B): p. 417-26.
- Hynes, R.O., The Extracellular Matrix: Not Just Pretty Fibrils. Science, 2009. 326(5957): p. 1216-1219.
- Muzzarelli, R., Chitins and chitosans for the repair of wounded skin, nerve, cartilage and bone. Carbohydrate Polymers, 2009. 76(2): p. 167-182.
- Ulubayram, K., Nur Cakar, A, Korkusuz, P, Ertan, C and Hasirci, N, EGF containing gelatin-based wound dressings. Biomaterials, 2001. 22(11): p. 1345-56.
- Estes, J., Scott Adzick, N, Harrison, MR, Longaker, MT and Stern, R, Hyaluronate metabolism undergoes and ontogenic transition during fetal development: Implications for Scar-free wound healing. Journal of Pediatric Surgery, 1993. 28(10): p. 1227-1231.
- Priyadarshi, R.a.R., J, Chitosan-based biodegradable functional films for food packaging applications. Innovative Food Science & Emerging Technologies, 2020. 62: p. 102346.
- Muxika, A., Etxabide, A, Uranga, J, Guerrero, P and de la Caba, K, Chitosan as a bioactive polymer: Processing, properties and applications. Int J Biol Macromol, 2017. 105(Pt 2): p. 1358-1368.
- Watanabe, J., Iwamoto, S and Ichikawa, S, Entrapment of some compounds into biocompatible nano-sized particles and their releasing properties. Colloids Surf B Biointerfaces, 2005. 42(2): p. 141-6.
- Younes, I., Rinaudo, M, Chitin and chitosan preparation from marine sources. Structure, properties and applications. Mar Drugs, 2015. 13(3): p. 1133-74.
- Bharadwaz, A., Jayasuriya, AC, Recent trends in the application of widely used natural and synthetic polymer nanocomposites in bone tissue regeneration. Materials Science and Engineering: C, 2020. 110: p. 110698.
- Kavya, K., Jayakumar, R, Nair, S and Chennazhi, KP, Fabrication and characterization of chitosan/gelatin/nSiO2 composite scaffold for bone tissue engineering. Int J Biol Macromol, 2013. 59: p. 255-63.
- Mezzana, P., Clinical efficacy of a new chitin nanofibrils-based gel in wound healing. Acta Chir Plast, 2008. 50(3): p. 81-4.
- Muzzarelli, R., Mattioli-Belmonte, M, Pugnaloni, A and Biagini, G, Biochemistry, histology and clinical uses of chitins and chitosans in wound healing. EXS, 1999. 87: p. 251-64.
- Sashiwa, H., Saimoto, H, Shigemasa, Y, Ogawa, R and Tokura, S, Lysozyme susceptibility of partially deacetylated chitin. Int J Biol Macromol, 1990. 12(5): p. 295-6.
- Jayakumar, R., Prabaharan, M, Sudheesh Kumar, PT, Nair, SV and Tamura, H, Biomaterials based on chitin and chitosan in wound dressing applications. Biotechnol Adv, 2011. 29(3): p. 322-37.
- Hamedi, H., Moradi, S, Hudson, SM and Tonelli, AE, Chitosan based hydrogels and their applications for drug delivery in wound dressings: A review. Carbohydr Polym, 2018. 199: p. 445-460.
- Anal, A., Stevens, W, Chitosan-alginate multilayer beads for controlled release of ampicillin. Int J Pharm, 2005. 290(1-2): p. 45-54.
- Anal, A., Tobiassen, A, Flanagan, J and Singh, H, Preparation and characterization of nanoparticles formed by chitosan-caseinate interactions. Colloids Surf B Biointerfaces, 2008. 64(1): p. 104-10.
- Berger, J., Reist, M, Mayer, JM, Felt, O, Peppas, NA and Gurny, R, Structure and interactions in covalently and ionically crosslinked chitosan hydrogels for biomedical applications. Eur J Pharm Biopharm, 2004. 57(1): p. 19-34.
- Maged, A., Abdelkhalek, AA, Mahmoud, AA, Salah, S, Ammar, MM and Ghorab, MM, Mesenchymal stem cells associated with chitosan scaffolds loaded with rosuvastatin to improve wound healing. Eur J Pharm Sci, 2019. 127: p. 185-198.
- Masson-Meyers, D., Andrade, TAM, Caetano, GF, Guimaraes, FR, Leite, MN, Leite, SN, Frade, MAC, Experimental models and methods for cutaneous wound healing assessment. Int J Exp Pathol, 2020. 101(1-2): p. 21-37.
- Wong, V., Sorkin, M, Glotzbach, JP, Longaker, MT, Gurtner, GC, Surgical approaches to create murine models of human wound healing. J Biomed Biotechnol, 2011. 2011: p. 969618.
- Peplow, P., Chung, TY, Baxter, GD, Laser photobiomodulation of proliferation of cells in culture: a review of human and animal studies. Photomed Laser Surg, 2010. 28 Suppl 1: p. S3-40.
- Papier, A., Peres, MR, Bobrow, M, Bhatia, A, The digital imaging system and dermatology. Int J Dermatol, 2000. 39(8): p. 561-75.
- Geer, D., Swartz, DD, Andreadis, ST, In vivo model of wound healing based on transplanted tissue-engineered skin. Tissue Eng, 2004. 10(7-8): p. 1006-17.
- Ibeh, B., Experimental Animal Models of Human Diseases - An Effective Therapeutic Strategynull2018: IntechOpen.
- Nagar, H., Srivastava, AK, Srivastava, R, Kurmi, ML, Chandel, HS, Ranawat, MS, Pharmacological Investigation of the Wound Healing Activity of <i>Cestrum nocturnum</i> (L.) Ointment in Wistar Albino Rats. Journal of Pharmaceutics, 2016. 2016: p. 9249040.
- Sirohi, B., Sagar, R, Effect of Hydroalcoholic Extract of Dactylorhiza Hatagirea Roots & Lavandula Stoechas Flower on Thiopental Sodium Induced Hypnosis in Mice. Journal of Drug Delivery and Therapeutics, 2019. 9: p. 414-417.
- El-Bahy, A., Aboulmagd, YM and Zaki, M, Diabetex: A novel approach for diabetic wound healing. Life Sci, 2018. 207: p. 332-339.

Round 2
Reviewer 2 Report
The authors of the manuscript responded to the submitted reviewer's comments - therefore I recommend to accept the work in its present form.